# What We Learn from Surveillance of Microbial Colonization in Recipients of Pediatric Hematopoietic Stem Cell Transplantation

**DOI:** 10.3390/antibiotics12010002

**Published:** 2022-12-20

**Authors:** Gabriele Kropshofer, Benjamin Hetzer, Miriam Knoll, Andreas Meryk, Christina Salvador, Evelyn Rabensteiner, Roman Crazzolara

**Affiliations:** 1Department of Pediatrics, Medical University of Innsbruck, 6020 Innsbruck, Austria; 2Institute of Hygiene and Medical Microbiology, Medical University of Innsbruck, 6020 Innsbruck, Austria

**Keywords:** surveillance, microbial colonization, pediatric hematopoietic stem cell transplant, antibiotic resistance

## Abstract

Infections in hematopoietic stem cell transplant (HSCT) remain one of the major causes for morbidity and mortality, and it is still unclear whether knowledge of microbial colonization is important. In this single-center study, we collected weekly surveillance cultures in pediatric recipients of allogenic HSCT from five different body regions and tested for bacteria and fungi. Between January 2010 and December 2021, we collected 1095 swabs from 57 recipients of allogeneic HSCTs (median age: 7.5 years, IQR 1–3: 2.5–11.9). The incidence of positive microbiological cultures (*n* = 220; 20.1%) differed according to the anatomic localization (*p* < 0.001) and was most frequent in the anal region (*n* = 98), followed by the genital, pharyngeal and nasal regions (*n* = 55, *n* = 37 and *n* = 16, respectively). Gram-positive bacteria (70.4%) were the most commonly isolated organisms, followed by fungi (18.6%), Gram-negative (5.5%), non-fermenting bacteria (1.4%), and other flora (4.1%). No association with increased risk of infection (*n* = 32) or septicemia (*n* = 7) was noted. Over time, we did not observe any increase in bacterial resistance. We conclude that there is no benefit to surveillance of microbial colonization by culture-based techniques in pediatric HSCT. Sequencing methods might enhance the detection of pathogens, but its role is still to be defined.

## 1. Introduction

The occurrence of severe infections are one of the major challenges following hematopoietic stem cell transplant (HSCT), as they are associated with high morbidity and mortality rates [1,2]. The unquestioned use of broad-spectrum antibiotics results in the development of antimicrobial resistance (AMR), which is estimated as a major cause of death worldwide [3,4]. Close monitoring of resistance is essential, as it is crucial to prevent lethal outcomes in the future [5].

HSCT is one of the most successful treatments in both malignant and non-malignant disease and involves the administration of healthy hematopoietic stem cells to compromised bone marrow. Annual reports of approximately 45,000 HSCTs in Europe and 5-year survival rates of more than 70% document its importance [6,7]. Contrastingly, treatment intensification limits its success, since HSCT results in high toxicity rates and pressures clinical research to identify better treatment strategies [8].

In the early and intermediate post-transplant phase, patients develop severe immunodeficiency and show few signs of infection [9]. Knowledge of microbiological colonization in HSCT recipients may be of major importance for the management of infections. In fact, most transplant centers have installed standard operating procedures (SOPs) for the development of complex surveillance programs to early detect colonization with fungi and bacteria and initiate preemptive strategies [10,11]. In the case of pathogen detection, antimicrobial susceptibility testing is performed to direct antibiotic treatment. Whether this strategy is useful for assessing colonization and analyzing resistance patterns for guiding the treatment of infections has not yet been adequately studied. Importantly, the great effort and number of resources expended on this issue are remarkable [12,13,14,15,16,17].

Following our SOPs, we obtained weekly surveillance cultures from five different body regions of all pediatric patients undergoing HSCT during a period of 12 years in a single center. We investigated the association of routine surveillance cultures with the development of infections in the most vulnerable phase of HSCT, which is the phase between neutropenia until neutrophil engraftment. Finally, we analyzed the shifts in microbial flora over time, in order to detect the emergence of resistant strains.

## 2. Material and Methods

The Ethics Committee of the Medical University of Innsbruck approved this study (EC No. 1301/2020) and waived the need for patient consent because of its retrospective nature. We performed this study in accordance with the Declaration of Helsinki.

### 2.1. Patients

In this study, we retrospectively collected the medical records of 57 recipients of allogeneic HSCTs at the Department of Pediatrics of the Medical University of Innsbruck between 1 January 2010 and 31 December 2021. We re-transplanted three patients during the study period and viewed them as new cases (60 cases in total). We performed transplants in patients for both malignant and non-malignant disease. Data included baseline demographics, baseline pathology, microbiological diagnostics and therapeutic treatments, as well as the number and type of complications, and were transferred to electronic records.

### 2.2. Experimental Design

Management of infections in patients after HSCT included prophylaxis, preemptive therapy and targeted therapies, and followed guidelines from evidence-based recommendations of the EBMT working group [18,19]. As per institutional standard, surveillance cultures were taken once weekly (on Mondays) from five different body areas. Results from positive cultures, as well as resistance testing, were considered for choice or modification of empirical therapy. For the purpose of this study, data was collected retrospectively, and epidemiology and AMR was finally analyzed.

### 2.3. Transplant Procedure and Supportive Care

Before starting patient conditioning, a 2-lumen tunneled central venous catheter (CVC) was inserted. During transplant, patients were admitted to a single bed laminar airflow room (LAF, HEPA filtration) from the beginning of conditioning until neutrophil recovery. Hand hygiene compliance rate as defined by the WHO guidelines was 83%, and the nurse/patient ratio was 1:2. Supportive care consisted of *Pneumocystis Jirovecii* prophylaxis with trimethoprim-sulfamethoxazole given at 5 mg/kg/bid ten days before transplantation and was restarted after neutrophil engraftment >1.0 × 10^9^/L (“take”). The antifungal prophylaxis regimen consisted of liposomal amphotericin-B given at 3–5 mg/kg three times a week. No systemic antibacterial prophylaxis was given during the neutropenic phase, nor was total gut decontamination performed. Cytomegalovirus (CMV) infection or reactivation prophylaxis consisted of acyclovir given at 10 mg/kg/tid at least until day +30. Patients at risk for CMV received ganciclovir. Irradiated (15 Gray) or inactivated, leukocyte-depleted red cell and single-donor platelet transfusions were used to maintain hemoglobin levels above 7.0 g/dL and the platelet number above 10 × 10^9^/L.

Barrier precautions were fulfilled according to international transplantation guidelines [1]. In particular, patients did not receive neutropenic food. Unpasteurized milk and milk products were avoided. Furthermore, the four essential steps “clean, separate, cook and chill” were strictly complied with [20].

### 2.4. Infections

Fever of unknown origin (FUO) was defined as a single temperature peak over 38.5 °C without any other explanation and/or a microbiologically documented infection (septicemia). In the case of FUO, at least two quantitative blood cultures were obtained and preemptive treatment was started. In patients with a first fever episode of ≥38.5 °C, treatment consisted of meropenem (20 mg/kg/tid) and gentamicin (5 mg/kg/d). Vancomycin (20 mg/kg/bid) was added within 24 h.

### 2.5. Surveillance Cultures

Species identification and susceptibility testing were performed at the Institute for Hygiene and Medical Microbiology of the Medical University of Innsbruck. Surveillance cultures were obtained from the right and left side of the nose, and the throat, genital and anal region on admission and weekly thereafter until engraftment. On clinical suspicion, swabs were taken from additional locations (listed as other). All samples were tested for bacteria and yeast using standard microbiological techniques. Cotton swabs from body sites were used to inoculate one Columbia Broth (Becton Dickinson, Heidelberg, Germany) and one Columbia Blood Agar (Becton Dickinson, Heidelberg, Germany) culture medium. Cultures were incubated at 37 °C and analyzed after 24 and 48 h.

### 2.6. Blood Culture Testing

Upon clinical suspicion of systemic infection, at least two blood samples were injected into BD BACTEC™ *Peds Plus* TM/*F* culture vials (enriched Soybean Casein Digest Broth with CO_2_) and upon arrival at the lab immediately processed in the BACTEC FX (Becton Dickinson, Heidelberg, Germany) blood culture system. Blood cultures were incubated for five days. On positivity, Gram-staining and sub-cultivation on agar plates were performed according to the standard techniques [21]. Positive samples were cultivated on Columbia Blood Broth, chocolate, MacConkey and Schaedler Anaerobic Agar culture media (all Becton Dickinson, Heidelberg, Germany) and incubated for 24 h at 37 °C in aerobic and 48 h in anaerobic conditions.

Species identification of positive cultures was performed by matrix-assisted laser desorption/ionization time of flight mass spectrometry (MALDI-TOF MS, Bruker Daltonik, Bremen, Germany) using the reference Biotyper library v4.1 (Bruker Daltonik, Bremen, Germany). Antimicrobial susceptibility testing was performed according to NCCLS/CLSI guidelines until 2011 [22]. In 2011, Austrian microbiological laboratories switched their methodology to EUCAST (breakpoint tables for interpretation of MICs and zone diameters—2011–2021, Versions 1.3 to 11.0). Strains were classified as susceptible or resistant according to the breakpoints applied in the year of their isolation.

### 2.7. Statistical Analysis

Descriptive statistics were performed for all variables of interest, giving medians and interquartile ranges for quantitative variables, and absolute and relative frequencies for qualitative variables. The chi-square test was applied to analyze the anatomical localization of positive cultures. Differences were considered statistically significant at *p* < 0.05. Data visualization and analysis was performed using the IBM Statistical Package for the Social Sciences (SPSS^®^), version 24 (IBM Corp., Armonk, NY, USA).

## 3. Results

### 3.1. Patient Characteristics

Patient characteristics are listed in Table 1.

Altogether, 60 allogeneic HSCTs were performed in 57 pediatric patients between 1 January 2010 and 31 December 2021. Three patients received two transplants. Of all the transplants, 39 (65%) were male and 21 (35%) were female. The median age was 7.5 years (IQR 1–3: 2.5–11.9).

The underlying diseases included leukemia (acute lymphoblastic and myeloid, *n* = 39), non-malignant hematologic disorders (thalassemia, sickle cell disease, severe aplastic anemia, *n* = 11), immunodeficiency syndromes (Wiskott–Aldrich syndrome, severe combined immunodeficiency, *n* = 4) and others (lymphoma, hemophagocytosis, solid tumor, erythropoietic protoporphyria; *n* = 6). The majority of cases (*n* = 46; 76.7%) received a total body irradiation (TBI; 12 Gray) containing a conditioning regimen, while the others received myeloablative chemotherapy alone before stem cell transplant. Graft-versus-host disease (GvHD) prophylaxis usually consisted of cyclosporine 1.5 mg/kg/bid in combination with methotrexate 10 mg/m² on days +1, +3 and +6. Thirty-four (56.7%) patients received a graft from a matched unrelated donor (MUD), while 18 (30%) patients received a matched sibling transplant (MSD), five (8.3%) patients were grafted from a haploidentical family donor and three (5%) patients from a mismatched unrelated donor (MMUD). The source of stem cells was un-manipulated bone marrow (70%) or peripheral blood stem cells (30%). Median time to engraftment was 19.5 days (IQ1–IQ3, 16–24 days).

### 3.2. Surveillance Cultures Obtained from Patients

Serial surveillance cultures (*n* = 1095) were obtained from a maximum of five different body sites for each of the 60 transplants (Figure 1).

In total, 79.9% of the swabs remained sterile. The most commonly cultured bacteria from positive swabs were *Enterococcus* spp. (*n* = 100, 44.8%), Coagulase-negative staphylococci (CoNS) (*n* = 53, 23.8%) and *Candida* spp. (*n* = 41, 18.4%). All isolates of *Enterococcus* spp. showed the expected phenotype, displaying no exceptional resistance patterns. Of the CoNS, only three showed no acquired resistances to antibiotics, whereas the rest showed varying combinations of resistances, most commonly to trimethoprim-sulfamethoxazole (90.6%), macrolides (80.1%), methicillin (67.9%), clindamycin and quinolones (39.6% each). Other less common resistances included fosfomycin, fusidic acid, tetracycline, aminoglycosides and rifampicin. Nine out of twelve *Enterobacterales* were *E. coli* and, while most isolates showed an expected phenotype or resistance only to ampicillin and trimethoprim-sulfonamide, there were two extended-spectrum beta-lactamase (ESBL) producers. One isolate of *Pseudomonas aeruginosa* was resistant to carbapenems, however without additional phenotypic resistances.

The region with the largest number of positive cultures was the anal area (*n* = 98, 44.5%) with a predominance of *Enterococcus* spp. (*n* = 65, 66.3%), followed by *Candida* spp. (*n* = 13, 13.3%), CoNS (*n* = 13, 13.3%) and *Enterobacterales* (*n* = 4, 4.1%). The second most colonized body region was the genital area (*n* = 55, 25%) with a prevalence of *Enterococcus* spp. (*n* = 29, 52.7%), followed by *Candida* spp. and CoNS (*n* = 9, 16.4% each), and *Enterobacterales* found in six (10.9%) patients. Altogether, 37 swabs from the throat region showed positive results, with a high percentage of *Candida* spp. (*n* = 16, 43.2%), followed by positive results for CoNS (*n* = 8, 21.6%), *Enterococcus* spp. (*n* = 5, 13.5%), *Enterobacterales* (*n* = 2, 5.4%) and other flora (*n* = 4, 10.8%). Sixteen positive nasal swabs showed colonization with mainly CoNS (*n* = 10, 62.5%), followed by *S. aureus*, *S. pneumoniae* (*n* = 1, 6.3%) and other flora.

### 3.3. Infections

In the 60 allogeneic cases of HSCT, 32 patients (53.3%) with FUO episodes occurred in the predefined study period from the induction of aplasia to neutrophil engraftment. Six patients were diagnosed with septicemia and one patient suffered a first COVID-19 infection. *Corynebact.* spp. and *Granulicatella adiacens* were detected in the blood culture of only one patient; *Acinetobacter baumanii* and *Pseudomonas aeruginosa* were isolated in a different patient. The other patients showed positive blood culture results for *Fusobacterium nucleatum*, *Streptococcus mitis*, *Staphylococcus epidermidis* and *Pseudomonas aeruginosa*. No correlation was found between body contamination and the pathogens detected in blood cultures from any of the included patients.

### 3.4. Mortality

None (*n* = 0) of the patients died during the pre-engraftment phase. Long-time survival was 70.2% (*n* = 40). Thirteen (22.8%) patients died because of disease recurrence, the mortality of three patients (5.3%) was due to GvHD and/or veno-occlusive disease (VOD), and one (1.7%) patient developed encephalitis associated with hemophagocytic lymphohistiocytosis.

## 4. Discussion

Infections are major complications in patients undergoing HSCT, and are the most important risk factor for transplant-related morbidity and mortality [23]. The pathophysiologic mechanisms include effects of previous high-dose cytotoxic therapy on physical integrity, consisting of mucositis and hemorrhagic cystitis, and impairment of the immune system, including prolonged neutropenia and immunosuppressive treatment. In particular, the insertion of central venous catheters (CVCs) may increase the risk for infections, as well as the widespread use of antimicrobial prophylaxis enhancing the development of resistance [24,25]. In fact, the skin harbors millions of microorganisms (bacteria, fungi, viruses) living in symbiotic relationship with their host [26] and which are critically involved in protecting against invading pathogens and the training of the immune system [27,28]. In addition, the gut microbiota play an essential role in regulating immune homeostasis, as they contain trillions of microorganisms, namely tenfold the number of cells in the human body. If the barriers including the skin or mucosa are disturbed, commensal but also pathogenic microorganisms can invade the human organism and cause severe infections [29]. For patients with high-dose myeloablative conditioning regimens, this may result in the development of mucositis, which implicates the disruption of the physiologic “protection wall”. As all patients receive a central catheter, the risk for pathogen entry is further increased. As a result, the two main sources of bacterial infection during the neutropenic phase are known to be infections caused by the normal endogenous gastrointestinal flora—mainly caused by Gram-negative bacteria—and CVC infections, predominantly caused by Gram-positive bacteria [25,30].

New studies are needed to assess microbial colonization and transplant management. Therefore, we retrospectively analyzed 60 pediatric allogeneic HSCTs and for whom weekly bacteriological body surveillance cultures were evaluated according to our standardized protocol. Patients did not receive antibacterial prophylaxis or undergo total gut decontamination since several studies have shown that these appear to provide no benefit [31,32]. The purpose of this retrospective analysis of the surveillance cultures was to detect the frequent types of bacteria colonizing the patients’ bodies, a possible shift in bacteria over time and the emergence of resistant strains. Finally, we analyzed the correlation between colonization and FUO/septicemia, predicting infections after SCT.

In our study, 79.9% of the 1095 swabs remained sterile. The presence of Gram-positive bacteria was confirmed and was within the expected spectrum of the physiologic bacterial flora [26]. However, the distribution of detected microorganisms differed significantly according to the region of origin. Several other studies have demonstrated that there is a huge topographical influence not only in the number, but also in the composition of commensal bacteria on the skin [33]. The predominance of CoNS in nasal swabs (62.5%), followed by *Staphylococci*, represents the typical flora in pediatric nasal microbiota [34]. Surprisingly, the most common microorganism in the oral cavity (throat) was *Candida* spp. In healthy persons, the most common microorganisms in this region are bacteria, but it is known that, especially in immunocompromised patients, they play a major role as opportunistic pathogens [35]. The largest number of positive surveillance cultures was seen in the genital and anal area with a high prevalence of *Enterococcus* spp., followed by *Candida* spp., CoNS and *Enterobacterales*. This composition is commonly seen in gut microbiota.

Regarding the impact of culture-based detection of colonization on different body sites for infections in neutropenic patients after HSCT, data is limited. The results of the studies are contradictory and subject to debate about the influence of bacterial commensal colonization. Dhaney et al. showed that colonization reported on weekly rectal swab surveillance cultures showed no correlation with clinical outcomes, and antimicrobial susceptibility testing reports did not correlate with in vivo clinical response [13]. On the other hand, oropharyngeal colonization without clinical signs of infection seemed to have clinical impact [36]. Heidenreich et al. showed that colonization with multidrug-resistant organisms (MDRO) did not show a negative impact on clinical outcome and patients were able to clear them after HSCT [37].

In this study, 6 (15.3%) of 39 febrile neutropenic episodes were attributed to blood stream infections. Two of those were detected by molecular methods only, while blood cultures remained negative. The cultured bloodstream pathogens showed no exceptional resistance, and the distribution of Gram-positive and Gram-negative infections was equal (50%/50%). Regarding the surveillance swabs, apart from two ESBL-producing *Enterobacterales* and one carbapenem-resistant *P. aeruginosa* isolate, no multiresistant strains were cultivated. Given the small sample number, no assumptions about the frequency or development over time were possible. Interestingly, however, the rate of resistance to trimethoprim-sulfamethoxazole in CoNS was higher than expected [38], possibly due to the previously applied *Pneumocystis Jirovecii* prophylaxis. None of the organisms detected by surveillance swab correlated with pathogens causing blood stream infections. Our study confirms the observation that pre-transplant colonization with bacteria or fungi does not correlate with increased risk for infections during the neutropenic phase.

We did not see any invasive fungal infection in our patient cohort, even though fungi species (18.6%) were the second most detected species in the surveillance cultures. Previously described mold-specific prophylaxis might be a reason why serious fungal infection was prevented [11,39].

The main limitations of this report are its retrospective nature and the inclusion of a limited number of patients in a single center. Furthermore, the effectiveness of traditional culture-based methods used in this study is questionable. Several studies have shown that culture-based methods underestimate the total diversity of the microorganism landscape due to different cultivation behaviors [26,40]. Consequently, it can therefore be assumed that the potential pathogens were not completely detected. Finally, sequencing technologies might significantly enhance the detection of potential pathogenic organisms, but a change in prophylactic therapies is unlikely.

## 5. Conclusions

Our analysis clearly shows that serial culture-based surveillance of body sites are not helpful in predicting infections or guiding antimicrobial treatment and therefore the value of this routine measurement in pediatric transplant recipients is not beneficial. Non-culture-based techniques, such as immunoassays, which detect microbial antigens or antibodies, or sequencing methods, which detect microbial RNA or DNA, might improve the sensitivity of detection. As this area will continue to expand, future studies are requested to evaluate the impact of colonization, but also its relevance in the prevention and management of infection after HSCT.

## Figures and Tables

**Figure 1 antibiotics-12-00002-f001:**
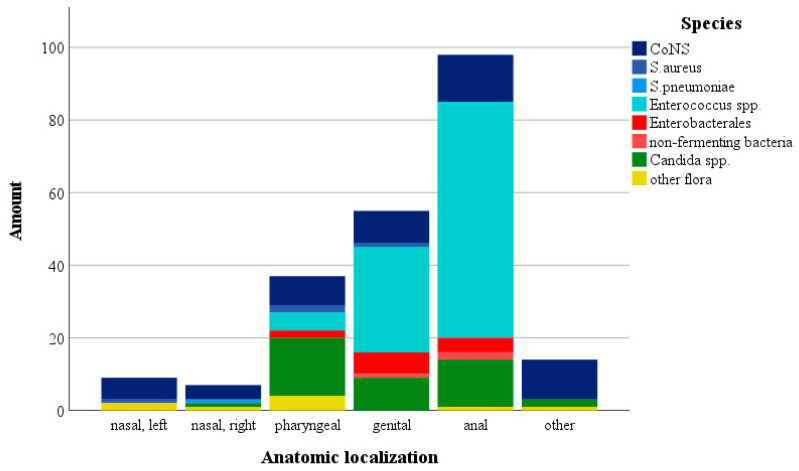
Distribution of serial surveillance cultures in different body sites. The amount of positive cultures and broken down by different strains and origin from the nasal (left + right), pharyngeal, genital, anal areas and other (umbilicus, insertion point of catheter) is indicated. Other flora included *Haemophilus influenzae* (*n* = 1), *Haemophilus parainfluenzae* (*n* = 1), *Streptococcus oralis* (*n* = 1), *Micrococcus luteus* (*n* = 2), *Streptococcus viridans* (*n* = 2) and *Corynebacterium* spp. (*n* = 2).

**Table 1 antibiotics-12-00002-t001:** Demographic and clinical characteristics of the study cohort.

Characteristics	Patient Data (*n* = 60) ^a^
Age, median (IQR), years	7.5 (2.5–11.9)
Sex	
female	21 (35.0)
male	39 (65.0)
Underlying diagnosis	
Leukemia	39 (65.0)
Hematologic	11 (18.3)
Immune deficiency	4 (6.7)
Others ^b^	6 (10.0)
Type of transplant	
MUD	34 (56.7)
MSD	18 (30.0)
Haploidentical	5 (8.3)
MMUD	3 (5.0)
Source of stem cells ^c^	
Bone marrow	42 (70.0)
Peripheral blood	16 (26.7)
Total body irradiation	
No	46 (76.7)
Yes	14 (23.3)
Engraftment, median (IQR), days	19.5 (16.0–24.0)

MUD: matched unrelated donor, MSD: HLA-matched sibling donor, MMUD: mismatched unrelated donor. ^a^ Unless indicated otherwise, data are expressed as No. (%) of patients. ^b^ Others include patients with the following diagnoses: two lymphoma, one Morbus Krabbe, one hemophagocytic lymphohistiocytosis, one erythropoietic protoporphyria and one Ewing sarcoma. ^c^ For two patients, information is missing.

## Data Availability

Not applicable.

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
