# Peer review of "What We Learn from Surveillance of Microbial Colonization in Recipients of Pediatric Hematopoietic Stem Cell Transplantation"

_antibiotics, 2022, doi:10.3390/antibiotics12010002_

Round 1

Reviewer 1 Report

The authors collected weekly surveillance cultures in pediatric recipients of allogenic HSCT from five different body regions and tested for bacteria and fungi. Culture-positive swabs were compared to positive blood cultures, and resistance patterns were analyzed. Between January 2010 and December 2021, we collected 1095 swabs from 57 recipients of allogeneic HSCTs (median age: 7.5 years, IQR 1-3: 2.5 – 11.9). The incidence of positive microbiological cultures (n=220; 20.1%) differed according to the anatomic localization (p<0.001) and was most frequent in the anal region (n=98), followed by the genital, pharyngeal and nasal regions (n=55, n=37 and n=16, respectively). Gram-positive bacteria (70.4%) were the most commonly isolated organisms, followed by fungi (18.6%), Gram-negative (5.5%), non-fermenting bacteria (1.4%), and other resident flora (4.1%). No association with increased risk of infection (n=32) or septicemia (n=7) was noted. Over time, we did not observe any increase in bacterial resistance. We conclude that there is no benefit to surveillance of microbial colonization in pediatric HSCT.

The paper will be ready for publication after major revision.

Please highlight your contributions in introduction.

The figure should be replotted and edited. Use times new romans fonts, clear colors, and 400 DPI resolution.

” Medical appropriateness concerning serial assessment of microbiota colonization for

prediction of infections is still a matter of debate.…………….”, Revise this paragraph.

The manuscript should start by a strong paragraph.

The format of references is not suitable for the journal.

Use Mendeley or Endnote to fix the references format.

The abstract should be rewritten to reflect the significance of the proposed work. The current abstract shows a lot of background information.

Conclusion: What are the advantages and disadvantages of this study compared to the existing studies in this area?

Experimental design should be discussed.

Did you face any problems or failed trials before doing stem cell transplantation?

Why did you increase the male cases than female? Is there a reasonable cause?

Which was the easier and applicable source of stem cells ,bone marrow or peripheral blood?

The inspiration of your work must further be highlighted. Some suggested recent literatures should add.

Add future works as bullets.

Looking and wishes for the revised version.

Author Response

1.) The paper will be ready for publication after major revision.

We appreciate the reviewer for taking time to carefully review the manuscript and give detailed and constructive comments, which have greatly helped to improve this paper. Below is our summarized point-by-point response to each comment.

2.) Please highlight your contributions in introduction. The manuscript should start by a strong paragraph. The abstract should be rewritten to reflect the significance of the proposed work. The current abstract shows a lot of background information. Experimental design should be discussed. Conclusion: What are the advantages and disadvantages of this study compared to the existing studies in this area? The inspiration of your work must further be highlighted. Some suggested recent literatures should add. Add future works as bullets.

The contributions of your study are now better highlighted in the introduction of the revised version of the manuscript. Also, the abstract reflects better the significance of the work in the revised version. Information on the experimental design are now discussed. Conclusion and inspiration are now provided more detailed. References have been added.

3.) The figure should be replotted and edited. Use times new romans fonts, clear colors, and 400 DPI resolution. The format of references is not suitable for the journal. Use Mendeley or Endnote to fix the references format.

The authors apologize for the inconvenience of a free format submission. Fonts, style, colors, resolution and reference style have now been revised according to the journal style of “Antibiotics”.

4.)” Medical appropriateness concerning serial assessment of microbiota colonization for prediction of infections is still a matter of debate.…………….”, Revise this paragraph.

We have now revised this sentence into “New studies are needed to assess microbial colonization and transplant management. Therefore, we…..”.

5.) Did you face any problems or failed trials before doing stem cell transplantation?

There was a clear indication for hematopoietic stem cell transplant. All patients were included in clinical studies and registries, the procedure followed the EBMT guidelines. There were no particular problems before undergoing stem cell transplant.

 6.) Why did you increase the male cases than female? Is there a reasonable cause?

We included all consecutive transplants without any selection. The largest group of transplant patients had childhood leukemia, who are known to be more male patients.

7.) Which was the easier and applicable source of stem cells, bone marrow or peripheral blood?

If available, bone marrow is the preferred choice for stem cells in children due to the lower risk for chronic GvHD. The proportion of 70% bone marrow origin in our study corresponds to the international guidelines.

Reviewer 2 Report

Kropshofer et al in the article titled `What we learn from surveillance of microbial colonization in recipients of pediatric hematopoietic stem cell transplatation` describe very relevant matter in pediatric studies.

The article is well organized and written and the tables and figure of good quality.

The topic is very relevant and interesting, but also as the authors stress, the patient number might not be sufficient to give the final conclusion of the statistical relevance. I also agree with the authors that probably more efficient and modernized techniques should be applied in order to identify other possible microorganisms of interest. As a summary, this could be a good first step in further and more detailed analyses of HSCT cases in terms of microbial colonization.

Author Response

We are grateful to the reviewer for taking time to carefully review the manuscript and give a detailed and constructive comment. Accordingly, we have revised the summary of our work and highlight what future studies should further address.

Reviewer 3 Report

The authors investigated the role of weekly surveillance cultures in the management of HSCT patients. Despite the limitation of being a single center study with low number of patients, the findings underline that periodic surveillance cultures are not useful in all settings. The manuscript can benefit from some revisions.

Major comment

More detailed description of the infection prevention policies in the wards would be useful. 

- Are all patients hospitalized in single bed rooms? 

- The compliance rate for hand hygiene?

- Nurse/patient ratio

-…

Minor comments

- Enterobacterales instead of Enterobacteriaceae

- Figure 1. Other resident flora? 

Author Response

We appreciate the reviewer for taking time to carefully review the manuscript and give detailed and constructive comments. Below is our point-by-point response to each comment.

1.) Major comment

More detailed description of the infection prevention policies in the wards would be useful. 

- Are all patients hospitalized in single bed rooms?

- The compliance rate for hand hygiene?

- Nurse/patient ratio

-…

Thanks for pointing out this important issue. All patients were hospitalized in single bed air laminar flow rooms, hand hygiene compliance rate as per WHO guidelines was 83% and the nurse/patient ratio was 1:2. This information has been added to the method section accordingly.

2.) Minor comments

- Enterobacterales instead of Enterobacteriaceae

We have revised this term correctly.

- Figure 1. Other resident flora? 

The information on “other resident flora” is now changed to “other flora” and is included in the figure legend.

Round 2

Reviewer 1 Report

Accept.